

# Does Twitter language reliably predict heart disease? A commentary on Eichstaedt et al. (2015a)

Nicholas J.L. Brown and James C. Coyne

University Medical Center, University of Groningen, Groningen, Netherlands

## ABSTRACT

We comment on *Eichstaedt et al.*'s (*2015a*) claim to have shown that language patterns among Twitter users, aggregated at the level of US counties, predicted county-level mortality rates from atherosclerotic heart disease (AHD), with "negative" language being associated with higher rates of death from AHD and "positive" language associated with lower rates. First, we examine some of Eichstaedt et al.'s apparent assumptions about the nature of AHD, as well as some issues related to the secondary analysis of online data and to considering counties as communities. Next, using the data files supplied by Eichstaedt et al., we reproduce their regression- and correlation-based models, substituting mortality from an alternative cause of death—namely, suicide—as the outcome variable, and observe that the purported associations between "negative" and "positive" language and mortality are reversed when suicide is used as the outcome variable. We identify numerous other conceptual and methodological limitations that call into question the robustness and generalizability of Eichstaedt et al.'s claims, even when these are based on the results of their ridge regression/machine learning model. We conclude that there is no good evidence that analyzing Twitter data in bulk in this way can add anything useful to our ability to understand geographical variation in AHD mortality rates.

## INTRODUCTION

*Eichstaedt et al. (2015a)* claimed to have demonstrated that language patterns among Twitter users, aggregated at the level of US counties, were predictive of mortality rates from atherosclerotic heart disease (AHD) in those counties, with "negative" language (expressing themes such as disengagement or negative relationships) being associated with higher rates of death from AHD and "positive" language (e.g., upbeat descriptions of social interactions or positive emotions) being associated with lower AHD mortality. Eichstaedt et al. examined a variety of measures to demonstrate the associations between Twitter language patterns and AHD, including (a) the frequency of usage of individual words associated with either positive or negative feelings or behaviors, (b) the tendency of Twitter users to discuss "positive" (e.g., skilled occupations) or "negative" (e.g., feelings of boredom) topics, and (c) an omnibus model incorporating all of their Twitter data, whose

Corresponding author
Nicholas J.L. Brown,
nicholasjlbrown@gmail.com

performance they compared with one using only "traditional" predictors such as health indicators and demographic and socioeconomic variables.

The claims made by *Eichstaedt et al. (2015a)* attracted considerable attention in the popular media (e.g., *Izadi, 2015*; *Jacobs, 2015*; *Singal, 2015*), with many of these articles being based in large part on the *Association for Psychological Science*'s (*2015*) own press release. However, a close examination of Eichstaedt et al.'s article and data appears to reveal a number of potential sources of distortion and bias in its assumptions about the nature of AHD, the use of Twitter data as a proxy for the socioemotional environment and people's health, and the use of counties as the unit of analysis. Some of these problems are immediately obvious from reading Eichstaedt et al.'s article, while others only manifested themselves in the testing of the relevant data that we undertook.

Here, we first present a selection of the main problems that we identified when reading *Eichstaedt et al.*'s (*2015a*) article. We then expose further problems that emerged after we scrutinized the original data and also cross-validated the application of Eichstaedt et al.'s models to a different mortality outcome variable, namely suicide. We end with a discussion of some broader implications for the use of large-scale sources of data to draw conclusions about health and human behavior based on sophisticated computer models.

## Issues related to the idea of psychological causes of AHD

Perhaps a large part of the appeal of *Eichstaedt et al.*'s (*2015a*) claims about the potential for community-level psychological factors—notably, those that purportedly lead Twitter users to either make aggressive or otherwise anti-social outbursts, or, conversely, express upbeat and positive sentiments—to somehow affect local rates of death from AHD is that these claims echo the common belief, repeated in the first sentence of Eichstaedt et al.'s abstract, that hostile or highly stressed individuals are more susceptible to cardiovascular problems. The notion that a loudmouthed, dominating, aggressive person is somehow more likely to suddenly drop dead from a heart attack is widespread in lay or pop psychology thinking (e.g., *Riggio, 2014*), perhaps at least in part because it provides some comfort to people on the receiving end of such individuals' behavior. Indeed, although Eichstaedt et al. did not use the term "Type A personality" in their article, this stereotype—characterized by a tendency towards aggression and a variety of other negative interpersonal characteristics, as well as greater susceptibility to cardiovascular problems—is a staple part of the popular culture surrounding the relation between mental and physical health (e.g., *Wilson, 2009*). Yet despite initial promising findings suggesting a link between Type A behavior pattern (TABP) and cardiac events and mortality in small samples (*Friedman & Rosenman, 1959*), an accumulation of evidence from more recent large-scale studies has consistently failed to show reliable evidence for such an association (*Kuper, Marmot & Hemingway, 2002*). Appearances of positive findings could be generated using a range of distress or negative affectivity variables (*Smith, 2006*). However, it was then recognized that negative affectivity could not readily be separated from a full range of antecedent and concurrent biological, psychological, and social factors. At best, negative affectivity is likely to be no more than a non-informative risk marker (*Ormel, Rosmalen & Farmer, 2004*), not a risk factor for

AHD. Its apparent predictive power is greatly diminished with better specification and measurement of confounds (*Smith, 2011*).

A recent large scale study of over 150,000 Australians (*Welsh et al., 2017*) provides a typical example of the problem. Significant associations between levels of psychological distress and incidence of ischemic heart disease (a superordinate category in the ICD-10 classification that includes AHD as a subcategory) decreased as adjustments were made for demographic and behavioral characteristics until measurement error and residual confounding seemed to account for any remaining association. As the authors put it, "A substantial part of the distress–IHD association is explained by confounding and functional limitations . . . . Emphasis should be on psychological distress as a marker of healthcare need and IHD risk, rather than a causative factor" (*Welsh et al., 2017*, p. 1).

In contrast to TABP, socioeconomic conditions have long been identified as playing a role in the development of AHD. For example, *Clark et al. (2009)* noted the importance of access to good-quality healthcare and early-life factors such as parental socioeconomic status. However, no county-level measure of either of those variables appeared in *Eichstaedt et al.*'s (*2015a*) model or data set.

## Issues related to the etiology of AHD

As the single most common ICD-10 cause of mortality in the United States, AHD might have seemed like a natural choice of outcome variable for a study such as that of *Eichstaedt et al. (2015a)*. However, it is important to take into account some aspects of the nature and course of AHD. It is a chronic disease that typically develops asymptomatically over decades. The first recognition of AHD often follows an event such as an acute myocardial infarction or some other sudden incident, reflecting the fact that the cumulative build-up of plaque over time has finally caused a blockage of the arteries (*Funk, Yurdagul Jr & Orr, 2012*) rather than any specific change in the immediately preceding time period. Indeed, disagreement among physicians as to whether the cause of death is AHD or some other cardiac event is common (*Mant et al., 2006*). A definitive post-mortem diagnosis of AHD may require an autopsy, yet the number of such procedures performed in the United States has halved in the past four decades (*Hoyert, 2011*).

In contrast to AHD, there is another cause of death for which the existence of an association with the victim's recent psychological state is widely accepted, namely suicide (*Franklin et al., 2017*; *O'Connor & Nock, 2014*). Although suicide can be the result of long-term mental health problems and other stressors, a person's psychological state in the months and days leading up to the point at which they take their own life clearly has a substantial degree of relevance to their decision. Hence, we might expect any county-level psychological factors that act directly on the health and welfare of members of the local community to be more closely reflected in the mortality statistics for suicide than those for a chronic disease such as AHD.

## Issues related to the secondary analysis of data collected online

In the introduction to their article, *Eichstaedt et al. (2015a)* invoked Google's use of search queries related to influenza, "providing earlier indication of disease spread than the

Centers for Disease Control and Prevention" (p. 160) as a (presumably positive) example of how "digital epidemiology can support faster response and deeper understanding of public-health threats than can traditional methods" (p. 160). However, both Google's project to infer a relation between searches for certain terms and the immediate prevalence of an acute infectious disease, and Eichstaedt et al.'s attempt to correlate certain kinds of communication with chronic cardiovascular conditions, suffer from the fundamental problem that they are attempting to extract some kind of signal from what may very well be a large amount of noise. In fact, before it was quietly shut down in August 2015, Google Flu Trends (GFT) failed—in some cases spectacularly—to correctly predict the seasonal spread of influenza in the United States on several occasions. As *Lazer et al. (2014)* put it, in their review of the problems that beset GFT, "The core challenge is that most big data that have received popular attention are not the output of instruments designed to produce valid and reliable data amenable for scientific analysis" (p. 1203)—a remark that clearly also applies to the notionally random selection of tweets that constitute the Twitter "Garden Hose" data, especially since, as we show below, this selection may not have been entirely random.

As well as the limitations noted in Eichstaedt et al.'s article (*2015a*) and Supplemental Material (*2015b*)—such as the difference in Twitter user density across counties, and the fact that 7% of tweets were mapped to the wrong county—we note that there is also a potential source of bias in the geographical coding of their Twitter data, namely the assumption that the users who provided enough information to allow their tweets to be associated with a county represented an unbiased sample of Twitter users in that county. This requires people of all ages, genders, socio-economic status levels, and ethnic backgrounds to be equally likely to either list their city and state in their profile, or to enable geotagging of their Tweets. However, it seems entirely plausible that certain categories of individuals might be more likely to self-censor their profile information than others (for example, it could be that people who do not wish to reveal their location are more or less restrained in their use of hostile language). Given that only 16% of tweets could be mapped to counties, any bias in this area could have substantial consequences.

## Issues associated with considering counties as communities

As *Eichstaedt et al. (2015a)* p. 160 themselves noted, "Counties are the smallest socioecological level for which most CDC health variables and US Census information are available." Thus, these authors seem to have acknowledged that their use of counties as their unit of analysis was driven principally by (eminently reasonable) practical constraints. However, Eichstaedt et al.'s subsequent interpretation of their results (e.g., "language expressed on Twitter revealed several community-level psychological characteristics that were significantly associated with heart-disease mortality risk," p. 164) requires that counties also constitute meaningful communities. Indeed, this interpretation also implies that any psychological mechanism that might account for the relation between the vocabulary of Twitter users and the health outcomes of the wider population within any given county works in a similar way across all counties. Yet there seem to be several reasons to question such assumptions. First, the size and population of US counties varies

widely; both their land areas (ranging from 1.999 to 144,504.789 square miles) and their 2010 populations (from 82 to 9,818,605) span five orders of magnitude. Second, across the United States, the political and economic importance of counties as a level of government (between the municipality and state level) is highly variable, being generally greater in the South (*Benton, 2002*, p. 7; *Haider-Merkel, 2009*, p. 723); indeed, in Connecticut, Rhode Island, and much of Massachusetts, counties have no governmental function whatsoever. Third, it is not clear that many Americans identify at all closely with their county as a source of community (*Kilgore, 2012*). Fourth, within counties, socioeconomic and other factors often vary enormously:

> [I]n New York County, New York, . . . neighborhoods range from the Upper East Side and SoHo to Harlem and Washington Heights. . . . [I]n San Mateo County, California, . . . neighborhoods range from the Woodside estates of Silicon Valley billionaires to the Redwood City bungalows of Mexican immigrants. (*Abrams & Fiorina, 2012*, p. 206)

Given such diversity in the scale and sociopolitical significance of counties, we find it difficult to conceive of a county-level factor, or set of factors, that might be associated with both Twitter language and AHD prevalence with any degree of consistency across the United States. *Eichstaedt et al.* (*2015a*, p. 166) cited two meta-analyses (*Leyland, 2005*; *Riva, Gauvin & Barnett, 2007*), which they claimed provided support for the idea that "the aggregated characteristics of communities . . . account for a significant portion of variation in health outcomes," but both of those meta-analyses were based predominantly on small and relatively homogeneously-sized geographical areas (and Leyland's study examined only health-related *behaviors*, not outcomes). The approximate population of each area in Leyland's study was 5,000, while *Riva, Gauvin & Barnett (2007)* reported a median area population of 8,600; compare these with Eichstaedt et al.'s mean county population of 206,000 and median of 78,000. As *Beyer, Schultz & Rushton* (*2008*, p. 40) put it, "The county often represents an area too large to use in determining true, local patterns of disease."

Working with aggregated data sets, such as *Eichstaedt et al.*'s (*2015a*) county-level data, also raises the question of whether specific data items can be aggregated in a meaningful way to create a collective "characteristic" (cf. *Subramanian et al., 2008*). For example, it is difficult to know what interpretation to place on the median individual household income of a county, especially one with a highly diverse population. It is also worth noting that, as reported by Eichstaedt et al. in their Supplemental Tables document (*2015c*), the "county-level" data for all of the variables that measure "county-level" health in their study (obesity, hypertension, diabetes, and smoking) are in fact statistical estimates derived from state-level data using "Bayesian multilevel modeling, multilevel logistic regression models, and a Markov Chain Monte Carlo simulation method" (p. DS7). However, Eichstaedt et al. provided no estimates of the possible errors or biases that the use of such techniques might introduce.

Based on the above observations, we decided to reproduce a number of *Eichstaedt et al.*'s (*2015a*) analyses, using their original data files in conjunction with other sources of information, to see whether the assumptions made by these authors stand up to a critical examination.

[1]An earlier version of the present article, available in preprint form at https://psyarxiv.com/dursw, stated that we had not been able to obtain *Eichstaedt et al.*'s (*2015a*) code because it had not been made available in the same OSF repository as the data. We are happy to acknowledge here that Eichstaedt and colleagues had in fact made their code available on the website of their Differential Language Analysis ToolKit (DLATK) software project, a fact that they have now documented in their recent preprint (*Eichstaedt et al., 2018*). We followed the installation instructions for DLATK and were able to reproduce the analyses described by *Eichstaedt et al.* (*2015a*, p. 161) under the heading of "Predictive models".

## METHOD

We were able to download all of *Eichstaedt et al.*'s (*2015a*) data files from the relevant Open Science Foundation (OSF) repository, and the majority of the analyses that follow are based on these data.[1] We did not have access to the original Twitter "Garden Hose" data set, so our analyses rely on the summaries of word usage provided by Eichstaedt et al. in their data files.

Next, we downloaded county-aggregated, age-adjusted mortality data for 2009 and 2010 for the ICD-10 category I25.1 (atherosclerotic heart disease) from the Centers for Disease Control and Prevention (CDC) online public health database, known as Wonder (http://wonder.cdc.gov/), in order to check that we could reproduce *Eichstaedt et al.*'s (*2015a*) data set exactly. We also downloaded comparable mortality data for the ICD-10 categories X60–X84, collectively labeled "Intentional self-harm"—in order to test the idea that suicide might be at least as well predicted by Twitter language as AHD—as well as the data for several other causes of death (including all-cause mortality) for comparison purposes. Finally, we obtained data about the precise geographical locations of counties from the US Census Bureau (http://www.census.gov/geo/www/gazetteer/files/Gaz_counties_national.zip). All of our statistical analyses were performed in R; a copy of the analysis code can be found at https://osf.io/g42dw.

## RESULTS

Using data downloaded from the CDC Wonder database, we were able to reproduce *Eichstaedt et al.*'s (*2015a*) principal data file, named countyoutcomes.csv, exactly. We were also able to reproduce Eichstaedt et al.'s reported numerical results from their article to within rounding error using our own R code (for the dictionary and topic language variables) or by running their modeling software in a virtual machine environment (for the analyses under the heading of "Predictive models").

In the following sections, we report a number of findings that we made when exploring *Eichstaedt et al.*'s (*2015a*) data set and the other data that we obtained. The order of these sections follows the same structure that we used in the introduction of the present article, showing how these findings relate to the concerns that we expressed earlier under three broad headings: (a) the use of mortality from AHD as the outcome variable; (b) the use of county-level aggregated data; and (c) the use of patterns of language in posts to Twitter as the principal predictor of the outcome.

### Variability in ICD-10 coding of cause of death

The validity of the mortality data from the CDC, and in particular the ICD-10 coding of the cause of death, is crucial to establishing the validity of *Eichstaedt et al.*'s (*2015a*) findings. Examination of the mortality figures from the counties included in Eichstaedt et al.'s study shows that death rates from AHD in 2009–2010 ranged from 13.4 per 100,000 people (1.55% of all recorded deaths) in East Baton Rouge Parish, LA to 185.0 per 100,000 people (26.11% of all recorded deaths) in Richmond County, NY. It is not clear that any county-level environmental, economic, or social factors that might contribute to

the development of AHD would be sufficient to explain why this condition—which, as Eichstaedt et al. noted, is the single most prevalent ICD-10 cause of mortality in the United States—appears to cause 13.8 times more deaths in one county than another. The two counties just mentioned are both among the 10% most populous counties in the US, with a total of 7,001 and 6,907 deaths, respectively, being recorded there in 2009–2010; this suggests that the large difference in recorded mortality from AHD is unlikely to be a statistical fluke due to a limited sample. In contrast, for deaths from cancer, the range of per-county mortality rates across the 1,347 counties included by Eichstaedt et al. runs from 102.4 to 326.7 per 100,000 people—a factor of 3.19 from lowest to highest, which is little different from the range for all-cause mortality (478.8 to 1,390.3 per 100,000 people, a factor of 2.90 from lowest to highest). Eichstaedt et al.'s acknowledgement that "the coding on death certificates may be inconsistent" (p. 166) would thus appear to be somewhat of an understatement, at least as far as AHD is concerned. Indeed, it seems possible that at least part of the variance in AHD mortality (whether this is to be explained by "community-level psychological characteristics" or some other mechanism) might be due, not to differences in the *actual* prevalence of AHD as the principal cause of death, but rather to variations in the propensity of local physicians to *certify* the cause of death as AHD (*McAllum, St. George & White, 2005*; *Messite & Stellman, 1996*; *Stausberg et al., 2006*). Researchers who intend to study AHD mortality using county-level data may wish to take this possibility into account (cf. *Roth et al., 2017*).

## Use of mortality from AHD as the outcome variable

The chronic nature of AHD implies that, to the extent that its prevalence may be affected by deleterious or protective lifestyle and social factors—including *Eichstaedt et al.*'s (*2015a*, p. 164) purported "indicators of community-level psychosocial health"—it will have been necessary for these factors to have exerted their effects over a long period. However, there are two potential sources of discrepancy between the current psychosocial state of a person's county of residence and its past effects on their health. First, the socioeconomic climate of an area can change substantially in less than a generation. The decline in the fortunes of the city of Detroit provides a recent dramatic example of this (*LeDuff, 2014*), but economic development can also bring rapid positive change to parts of a state within quite a short time. Second, individuals tend not to stay in one place; the county where someone spends his or her childhood may be a long way from where he or she ends up working, and possibly different again from where the same person is living when symptoms of AHD emerge later in life, which might be after retirement. Data from the *US Census Bureau (2011)* show that in 2010 approximately 3.5% of Americans moved either to another county in the same state, or to another state altogether, a figure that appears from an examination of comparable data from other years to be relatively constant over time. Thus, it seems likely that a substantial proportion of the people who die from AHD each year in any given county may have lived in one or more other counties during the decades when AHD was developing, and thus been exposed to different forms, both favorable and unfavorable, of Eichstaedt et al.'s purported community-level psychological characteristics during that period.

**Table 1** Correlations between self-harm and Twitter language measured by dictionaries.

| Language variable | r | p | 95% CI |
|---|---|---|---|
| Risk factors | | | |
| Anger | −0.169 | <.001 | [−0.238, −0.099] |
| Negative relationships | −0.095 | .010 | [−0.166, −0.023] |
| Negative emotions | −0.102 | .005 | [−0.173, −0.030] |
| Disengagement | 0.008 | .831 | [−0.064, 0.080] |
| Anxiety | −0.045 | .219 | [−0.117, 0.027] |
| Protective factors | | | |
| Positive relationships[a] | −0.001 | .976 | [−0.073, 0.071] |
| Positive emotions | 0.059 | .110 | [−0.013, 0.131] |
| Positive engagement | −0.031 | .393 | [−0.103, 0.041] |

**Notes.**
[a]Following *Eichstaedt et al. (2015a)*, the word *love* was removed from the dictionary for this variable. See discussion in the text.

We suggested earlier that if a county-level psychological factor was sufficiently strong to influence mortality from AHD, it might also be expected to influence local rates of suicide. We therefore examined the relation of the set of causes of death listed by the CDC as "self-harm" with Twitter language usage, using the procedures reported in the first subsections entitled "Language variables from Twitter" and "Statistical analysis" of *Eichstaedt et al.*'s (*2015a*, p. 161) Method section. Because of a limitation of the CDC Wonder database, whereby county-aggregated mortality data are only returned for any given county when at least 10 recorded deaths per year, on average, match the requested criteria for the period in question, data for self-harm were only available for 741 counties; however, these represented 89.9% of the population of Eichstaedt et al.'s set of 1,347 counties.

In the "Dictionaries" analysis, we found that mortality from self-harm was *negatively* correlated with all five "negative" language factors, with three of these correlations (for anger, negative-relationship, and negative-emotion words) being statistically significant at the .05 level (see our Table 1). That is, counties whose residents made greater use of negative language on Twitter had lower rates of suicide, or, to borrow *Eichstaedt et al.*'s (*2015a*, p. 162) words, use of negative language was "significantly protective" against self-harm; this statistical significance was unchanged when income and education were added as covariates. In a further contrast to AHD mortality, two of the three positive language factors (positive relations and positive emotions) were *positively* correlated with mortality from self-harm, although these correlations were not statistically significant at the conventional .05 level.

Next, we analyzed the relation between Twitter language and self-harm outcomes at the "Topics" level. Among the topics most highly correlated with increased risk of self-harm were those associated with spending time surrounded by nature (e.g., *grand*, *creek*, *hike*; r = .214, CI[2] = [.144, .281]), romantic love (e.g., *beautiful*, *love*, *girlfriend*; r = .176, CI = [.105, .245]), and positive evaluation of one's social situation (e.g., *family*, *friends*, *wonderful*; r = .175, CI = [.104, .244]). There were also topics of discussion that appeared to be strongly "protective" against the risk of self-harm, such as baseball (e.g., *game*, *Yankees*, *win*; r =

[2]We report 95% CIs here for consistency with *Eichstaedt et al. (2015a)*; however, given their use of a Bonferroni-corrected significance threshold, it could be argued that Eichstaedt et al. should have reported 99.9975% CIs.

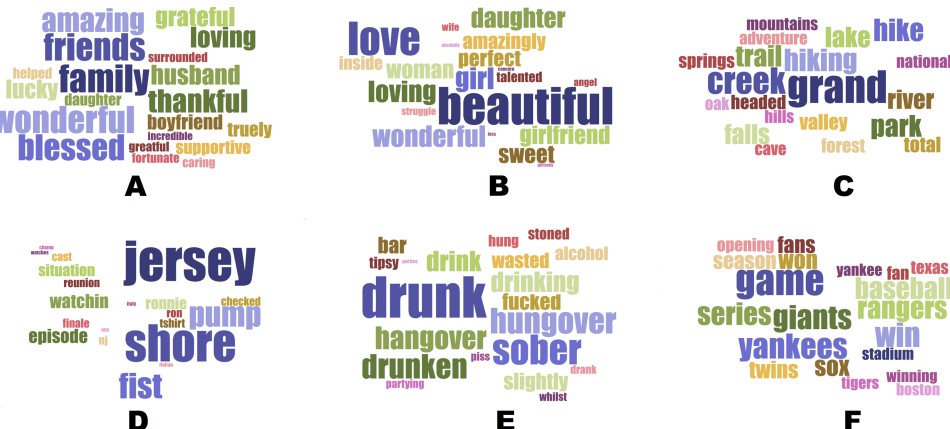

**Figure 1** **Twitter topics highly correlated with age-adjusted mortality from self-harm, cf.** *Eichstaedt et al.*'s *(2015a).* Figure 1 (A–C) Topics positively correlated with county-level self-harm mortality: (A) Friends and family, $r = .175$. (B) Romantic love, $r = .176$. (C) Time spent with nature, $r = .214$. (D–F) Topics negatively correlated with county-level self-harm mortality: (D) Watching reality TV, $r = -.200$. (E) Binge drinking, $r = -.249$. (F) Baseball, $r = -.317$.

$-.317$, CI $= [-.381, -.251]$), binge drinking (e.g., *drunk, sober, hungover*; $r = -.249$, CI $= [-.316, -.181]$), and watching reality TV (e.g., *Jersey, Shore, episode*; $r = -.200$, CI $= [-.269, -.130]$). All of the correlations between these topics and self-harm outcomes, both positive and negative, were significant at the same Bonferroni-corrected significance level (i.e., $.05/2,000 = .000025$) used by *Eichstaedt et al. (2015a)*, and remained significant at that level after adjusting for income and education. That is, several topics that were ostensibly associated with "positive," "eudaimonic" approaches to life predicted higher rates of county-level self-harm mortality, whereas apparently hedonistic topics were associated with lower rates of self-harm mortality, and the magnitude of these associations was at least as great—and in a few cases, even greater—than those found by Eichstaedt et al. These topics are shown in "word cloud" form (generated at https://www.jasondavies.com/wordcloud/) in our Fig. 1 (cf. *Eichstaedt et al.*'s *(2015a)* Figure 1).

This would seem to pose a problem for *Eichstaedt et al.*'s *(2015a,* p. 166) claim to have shown the existence of "community-level psychological factors that are important for the cardiovascular health of communities." Apparently the "positive" versions of these factors, while acting via some unspecified mechanism to make the community as a whole less susceptible to developing hardening of the arteries, also simultaneously manage to make the inhabitants of those counties more likely to commit suicide, and vice versa. We suggest that more research into the possible risks of increased levels of self-harm might be needed before "community-level psychological factors" were to be made the focus of intervention, as Eichstaedt et al. suggested in the final sentence of their article.[3]

## Bias caused by selection of counties

As noted above, the CDC Wonder database returns county-aggregated mortality data for any given cause of death only for those counties where at least 10 deaths from that cause were recorded per year, on average, during the period covered by the user's request. This cutoff

[3]In their recent preprint, *Eichstaedt et al. (2018)* noted that the relation between self-harm and positive Twitter language disappeared when they added measures of county-level rurality and elevation as covariates. We do not find this surprising. Our point is not that Twitter language actually predicts county-level mortality from suicide (or AHD); rather, it is that with a sufficient number of predictors, combined with unreliability in the measurement of these, one can easily find spurious relations between variables (cf. *Westfall & Yarkoni, 2016*).

means that *Eichstaedt et al.*'s (*2015a*) data set is skewed toward counties having higher rates of AHD. For example, AHD mortality was sufficiently high in Plumas County, CA (2010 population 20,007; 25 deaths from AHD in 2009–2010) for this county to be included, whereas the corresponding prevalence in McKinley County, NM (2010 population 71,492; 17 deaths from AHD in 2009–2010), as well as 130 other counties with higher populations than Plumas County but fewer than 20 deaths from AHD, was not. Thus, the selection of counties tends to include those with higher levels of the outcome variable, which has the potential to introduce selection bias (*Berk, 1983*). For example, counties with a 2010 population below the median (78,265) in Eichstaedt et al.'s data set had significantly higher AHD mortality rates than counties with larger populations (53.21 versus 49.87 per 100,000; $t(1344.6) = 3.12$, $p = .002$, $d = 0.17$). The effect of such bias on the rest of Eichstaedt et al.'s analyses is hard to estimate, but given that one of these authors' headline results—namely, that their Twitter-only model predicted AHD mortality "significantly better" than a "traditional" model, a claim deemed sufficiently important to be included in their abstract—had an associated $p$ value of .049, even a very small difference might be enough to tilt this result away from traditional statistical significance.

## Problems associated with county-level aggregation of data

We noted earlier that the diversity among counties made it difficult to imagine that the relation between Twitter language and AHD would be consistent across the entire United States. Indeed, when the counties in *Eichstaedt et al.*'s (*2015a*) data set are split into two equal subsets along the median latitude of their centroids (38.6484 degrees North, a line that runs approximately through the middle of the states of Missouri and Kansas), the purported effect of county-level Twitter language on AHD mortality as measured by Eichstaedt et al.'s dictionaries seems to become stronger in the northern half of the US than for the country as a whole, but mostly disappears in the southern half (see our Table 2). There does not appear to be an obvious theoretical explanation for this effect; if anything one might expect the opposite, in view of the observation made previously that counties may play a greater role in people's lives in the South.

A further issue with *Eichstaedt et al.*'s (*2015a*) use of data aggregated at the level of counties is that it resulted in an effective sample size that was much smaller than these authors suggested. For example, Eichstaedt et al. compared their model to "[t]raditional approaches for collecting psychosocial data from large representative samples . . . [which] are based on *only* [emphasis added] thousands of people" (p. 166). This suggests that these authors believed that their approach, using data that was generated by an unspecified (but implicitly very large) number of different Twitter users, resulted in a more representative data set than one built by examining the behaviors and health status of "only" a few thousand individuals. However, by aggregating their Twitter data at the county level, and merging it with other county-level health and demographic information, Eichstaedt et al. reduced each of their variables of interest to a single number for each county, regardless of that county's population. In effect, Eichstaedt et al.'s data set contains a sample of only 1,347 individual units of analysis, each of which has the same degree of influence on the conclusions of the study. A corollary of this is that, despite the apparently large

**Table 2** Partial correlations between atherosclerotic heart disease (AHD) mortality and Twitter language measured by dictionaries, across the northern and southern halves of the United States.

| Language variable | North | | | South | | |
|---|---|---|---|---|---|---|
| | Partial *r* | *p* | 95% CI | Partial *r* | *p* | 95% CI |
| Risk factors | | | | | | |
| Anger | 0.240 | <.001 | [0.168, 0.310] | −0.020 | .604 | [−0.095, 0.056] |
| Negative relationships | 0.156 | <.001 | [0.081, 0.229] | 0.060 | .121 | [−0.016, 0.135] |
| Negative emotions | 0.108 | .005 | [0.032, 0.182] | 0.028 | .462 | [−0.047, 0.104] |
| Disengagement | 0.166 | <.001 | [0.092, 0.239] | −0.012 | .750 | [−0.088, 0.063] |
| Anxiety | 0.017 | .654 | [−0.058, 0.093] | 0.104 | .007 | [0.028, 0.178] |
| Protective factors | | | | | | |
| Positive relationships[a] | −0.032 | .411 | [−0.107, 0.044] | 0.111 | .004 | [0.035, 0.185] |
| Positive emotions | −0.166 | <.001 | [−0.238, −0.091] | 0.082 | .034 | [0.006, 0.156] |
| Positive engagement | −0.192 | <.001 | [−0.264, −0.119] | 0.041 | .288 | [−0.035, 0.116] |

**Notes.**

Partial *r*: partial correlation coefficients obtained from a regression predicting AHD from the Twitter theme represented by the language variable, with county-level education and income as control variables.

[a]Following *Eichstaedt et al. (2015a)* the word *love* was removed from the dictionary for this variable. See discussion in the text.

number of participants overall, a very small group of voluble Twitter users could have a substantial influence in smaller counties. For example, in tweets originating from Jefferson County, WA (population 29,872) just nine instances of the word "fuck" (or derivatives thereof) appear in the entire data set, whereas the same word and its variants appear 9,271 times in the tweets of the residents of Montgomery County, NY (population 50,219), a per-inhabitant rate that is almost 600 times larger. Perhaps community standards of polite discourse vary rather more widely across rural areas of the United States than most people might imagine, but it seems at least equally likely that just a few angry people in mid-state New York are responsible for this avalanche of social media profanity, and that their input may consequently have had rather more impact on Eichstaedt et al.'s results than if those same few individuals had been living in Los Angeles County, CA (population 9,818,605). This aggregation into counties calls into question Eichstaedt et al.'s claim (p. 166) that an analysis of Twitter language can "generate estimates based on 10s of millions of people"; indeed, it could be that their results are being driven by just a few hundred particularly active Twitter users, particularly those living in smaller counties.[4]

## Apparent censorship of the Twitter data

An examination of the words and phrases that appear in *Eichstaedt et al.*'s (*2015a*) dictionaries suggests that some form of censorship may have been applied to the Twitter data. For example, the list of frequencies of each word in the dictionary contains entries for the words "nigga," "niggas," and "niggaz" (which, between them, appeared 1,391,815 times in the analyzed tweets, and at least once in all but five of the 1,347 included counties), but not "nigger" or "niggers." It seems highly unlikely that the most common spelling[5] of this word would not appear even once in any of the 148 million tweets that were included in Eichstaedt et al.'s analyses. A number of other common epithets for ethnic and religious groups are also absent from the dictionary–frequency table—although a search for such

[4]Examination of *Eichstaedt et al.*'s (*2015a*) data set shows that for 267 counties (19.8%), less than 100,000 words were included in the database, which corresponds to around 5,000 tweets. However, no indication is available of the number of unique Twitter users per county.

[5]The spelling "nigga" is often used by African Americans in a neutral or positive sense; the form "nigger" is the one typically used by members of other groups as a racial slur (*Goudet, 2013*).

words on Twitter suggests that they are in relatively common usage—as are the words "Jew[s]" and "Muslim[s]." In Appendix B of their recent preprint, *Eichstaedt et al. (2018)* have indicated that they were previously unaware of this issue, which we therefore presume reflects a decision by Twitter to bowdlerize the "Garden Hose" dataset. Such a decision clearly has substantial consequences for any attempt to infer a relation between the use of hostile language on Twitter and health outcomes, which requires that the tweets being analyzed are truly representative of the language being used. Indeed, it could be argued that there are few better examples of language that expresses interpersonal hostility than invective directed towards ethnic or religious groups.[6]

## Potential sources of bias in the "Topics" database

A further potential source of bias in *Eichstaedt et al.*'s (*2015a*) analyses, which these authors did not mention in their article or their supplemental documents (*Eichstaedt et al., 2015b*; *Eichstaedt et al., 2015c*), is that their "Topics" database was derived from posts on Facebook (i.e., not Twitter) by a completely different sample of users, as can be seen at the site from which we downloaded this database (http://wwbp.org/data.html). Furthermore, some of the topics that were highlighted by Eichstaedt et al. in the word clouds[7] in their Fig. 1 contain words that appear to directly contradict the topic label (e.g., "famous" and "lovers" in "Hate, Interpersonal Tension," left panel; "enemy" in "Skilled Occupations," middle panel; "painful" in "Positive Experiences," left panel). There are also many incorrectly spelled words, as in topic #135 ("can't, wait, afford, move, belive [*sic*], concentrate"), topic #215 ("wait, can't, till, tomorrow, meet, tomarrow [*sic*]"), topic #467 ("who's, guess, coming, thumbs, guy, whos [*sic*], idea, boss, pointing"), and many topics make little sense at all, such as #824 ("tooo, sooo, soooo, alll, sooooo, toooo, goood, meee, meeee, youuu, gooo, soooooo, allll, gooood, ohhh, ughhh, ohhhh, goooood, mee, soooooo") and #983 ("ur, urself, u'll, coz, u've, cos, urs, bcoz, wht, givin"). The extent to which these automatically extracted topics from Facebook really represent coherent psychological or social themes that might appear with any frequency in discussions on Twitter seems to be questionable, in view of the different demographics and writing styles on these two networks.

## Flexibility in interpretation of dictionary data

A problem for *Eichstaedt et al.*'s (*2015a*) general argument about the salutary effects of "positive" language was the fact that the use of words expressing positive relationships appeared, in these authors' initial analyses, to be positively correlated with AHD mortality. To address this, Eichstaedt et al. took the decision to eliminate the word *love* from their dictionary of positive-relationship words. Their justification for this was that "[r]eading through a random sample of tweets containing *love* revealed them to be mostly statements about loving things, not people" (p. 165). However, similar reasoning could be applied to many other dictionary words—including those that featured in results that did not contradict Eichstaedt et al.'s hypotheses—with the most notable among these being, naturally, *hate*. In fact, it turns out that *hate* dominated Eichstaedt et al.'s negative relationships dictionary (41.6% of all word occurrences) to an even greater degree than *love*

[6]We assume that the apparent omission of "Jew[s]" and "Muslim[s]" was motivated by concerns that at least some of the tweets mentioning these words might be expressing hostility towards these groups.

[7]It appears that the relative size of the words in *Eichstaedt et al.*'s (*2015a*, Fig. 1) word clouds is determined by the relative frequency of these words in the Facebook data from which the topics were derived, and does not represent the prevalence of these words in the Twitter data.

did for the positive relationships dictionary (35.8%). We therefore created an alternative version of the negative relationships dictionary, omitting the word *hate*, and found that, compared to the original, this version was far less likely to produce a statistically significant regression model when predicting AHD mortality (e.g., regressing AHD on negative relationships, controlling for income and education: with *hate* included, partial $r = .107$, $p <. 001$, 95% CI $= [.054, .159]$; with *hate* excluded, partial $r = -.005$, $p =. 849$, 95% CI $= [ -.057, .048]$). However, we do not have access to the full Twitter "Garden Hose" data set, and are thus unable to establish whether, analogously to what Eichstaedt et al. did for *love*, a similar examination of a "random sample of tweets" containing the word *hate* might "reveal" that they are mostly statements about hating things rather than people, thus providing an equivalent justification for dropping this word from the analyses.

In their Note 6, *Eichstaedt et al.* (*2015a*, p. 167) mentioned another justification for removing the word *love* from the positive relationships dictionary. They noted that "in lower-SES areas, users [may] share more about personal relationships on Twitter, which distorts the results obtained when using the original positive-relationships dictionary." Of course, it *might* be true of personal relationships, or indeed any other aspect of people's lives, that those who live in lower-SES areas—or, for that matter, those who are married, or smoke, or suffer from diabetes—tend to communicate more (or, indeed, less) about that topic on Twitter. But the factor analysis in Eichstaedt et al.'s Note 6 does not provide any direct evidence for their claim of a possible relation between residence in a lower SES area and a tendency to tweet about personal relationships.[8]

## Comparison of Twitter-based and "traditional" prediction models

*Eichstaedt et al.* (*2015a*, p. 161) reported that they "created a single model in which all of the word, phrase, dictionary, and topic frequencies were independent variables and the AHD mortality rate was the dependent variable." It is not clear exactly how this model was constructed or what weighting was given to the various components, even though the numbers of each category (words, phrases, dictionary entries, and topics) vary widely. In their 'Results', under the heading "Predictions," these authors compared the performance of what we might call their "Twitter omnibus" model with what they described as "traditional" models (i.e., those based on their demographic, socioeconomic, and health status variables), and claimed that the performance of the Twitter-based model was "significantly" better, based on *p* values of .026 and .049. However, the degrees of freedom here (1,346) are sufficiently numerous, and thus the statistical power to detect an effect sufficiently high, that these *p* values arguably constitute quite strong evidence *in favor* of the null hypothesis of no effect (cf. *Lakens & Evers, 2014*). It is also unclear from *Eichstaedt et al.*'s (*2015b*) Supplemental Material exactly how many predictors were finally included in their model.

### How similar are the comparative maps?

Figure 3 of *Eichstaedt et al.*'s (*2015a*) article shows two maps of the counties of the northeastern United States, with those counties that had sufficient cases of AHD mortality to be included in their sample being color-coded according to either CDC-reported or

[8] *Eichstaedt et al. (2018)* have recently explained that the exclusion of *love* from their positive relationships dictionary in their earlier article (*Eichstaedt et al., 2015a*) was the result of discussions with a reviewer of that article.

[9]More precisely, 200 counties (32.9%) had a discrepancy of three or more color intervals, while 123 counties (20.2%) had a discrepancy of six or more. Assuming for simplicity that rounding error is uniformly distributed, a difference of three intervals corresponds to a mean difference of 21.4 percentage points, and a difference of six intervals to a mean difference of 42.8 percentage points. Thus, even with the extremely conservative simplifying assumption that "three (six) or more color intervals" actually means "exactly three (six) intervals," the mean discrepancy across these counties is $((200 \times 21.4) + (123 \times 42.8)) / 323 = 29.5$ percentage points. Note also that the possible extent of the discrepancy is bounded at between seven and 13 color intervals, depending on the relative positions of the two counties along the 1–14 scale.

Twitter-predicted AHD mortality. *Eichstaedt et al.* (*2015a*, p. 164) claimed that "a high degree of agreement is evident" between the two maps. We set out to evaluate this claim by examining the color assigned to each county on each of the two maps, and determining the degree of difference between the mortality rates corresponding to those colors. To this end, we wrote a program to extract the colors of each pixel across the two maps, convert these colors to the corresponding range of AHD mortality rates, and draw a new map that highlights the differences between these rates using a simple color scheme. The color-based scale shown at the bottom of Eichstaedt et al.'s Figure 3 seems to imply that the maps are composed of 10 different color codes, each representing a decile of per-county AHD mortality, but in fact this scale is somewhat misleading. In fact, 14 different colors (plus white) are used for the counties in the maps in Eichstaedt et al.'s Figure 3, with each color apparently (assuming that each color corresponds to an equally-sized interval) representing around seven percentile rank places, or what we might call a "quattuordecile," of the AHD mortality distribution. For more than half (323 out of 608, or 53.1%) of the counties that have a color other than white in Eichstaedt et al.'s maps, the difference between the two maps is three or more of these seven-point "color intervals," as shown in our Fig. 2. Within these counties, the mean discrepancy is at least 29.5 percentage points, and probably considerably higher.[9] It is therefore questionable whether it is really the case that "[a] high degree of agreement is evident" (*Eichstaedt et al., 2015a*, p. 164) between the two maps, such that one might use the Twitter-derived value to predict AHD mortality for any practical purpose.

## DISCUSSION

In the preceding paragraphs, we have examined a number of aspects of *Eichstaedt et al.*'s (*2015a*) claims about the county-level effects of Twitter language on AHD mortality, using for the most part these authors' own data. We have shown that many of these claims are, at the least, open to alternative interpretations. The coding of their outcome variable (mortality from AHD) is subject to very substantial variability; the process that selects counties for inclusion is biased; the same regression and correlation models "predict" suicide at least as well as AHD mortality but with almost opposite results (in terms of the valence of language predicting positive or negative outcomes) to those found by Eichstaedt et al.; the Twitter-based dictionaries appear not to be a faithful summary of the words that were actually typed by users; arbitrary choices were apparently made in some of the dictionary-based analyses; the topics database was derived from a completely different sample of participants who were using Facebook, not Twitter; there are numerous problems associated with the use of counties as the unit of analysis; and the predictive power of the model, including the associated maps, appears to be questionable. While we were able to reproduce—at a purely computational level—the results of Eichstaedt et al.'s advanced prediction model, based on ridge regression and *k*-fold cross-validation, we do not believe that this model can address the problems of validity and reliability posed by the majority of the points just mentioned. In summary, the evidence for the existence of community-level psychological factors that determine AHD mortality better than traditional socioeconomic

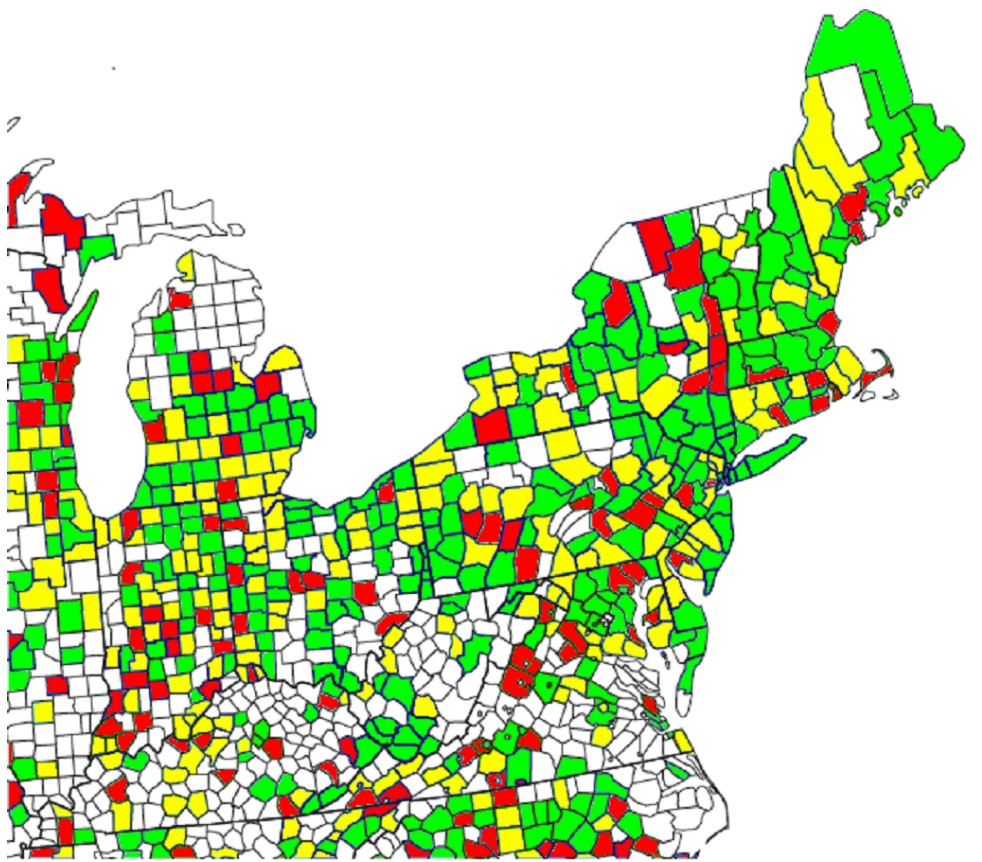

**Figure 2  Difference between the two maps of AHD mortality rates (CDC-reported and Twitter-predicted) from *Eichstaedt et al.*'s (*2015a*) Figure 3.** Note: Green indicates a difference of 0–2 color-scale points (see discussion in the text) between the two maps; yellow, a difference of 3–5 points; red, a difference of 6 or more points. Of the 608 colored areas, 123 (20.2%) are red and 200 (32.9%) are yellow.

and demographic predictors seems to be considerably less strong than Eichstaedt et al. claimed.

A tourist, or indeed a field anthropologist, driving through Jackson County (2010 population 42,376) and Clay County (2010 population 26,890) in southern Indiana might not notice very much difference between the two. According to the US Census Bureau (https://factfinder.census.gov/) these counties have comparable population densities, ethnic make-ups, and median household incomes. It seems unlikely that there would be a large variation in the "norms, social connectedness, perceived safety, and environmental stress, that contribute to health and disease" (*Eichstaedt et al., 2015a*, pp. 159–160) between these two rural Midwestern communities. Yet according to Eichstaedt et al.'s data, the levels of anger and anxiety expressed on Twitter were more than 12 and six times, respectively, higher in Jackson County than in Clay County. These differences are not easy to explain; any community-level psychological characteristics that might be driving them must obey some strange properties. Such characteristics would have to operate, at least partly, in ways that are not accounted for by variables for which Eichstaedt et al. applied statistical

controls (such as income and smoking prevalence), yet presumably they must have some physical manifestation in order to be able to have an effect on people's expressed feelings. It is difficult to imagine how such a characteristic might have gone unnoticed until now, yet be able to cause people living in very similar socioeconomic conditions less than 100 miles apart in the same state to express such varying degrees of negative emotionality. A more parsimonious explanation, perhaps, is that there is a very large amount of noise in the measures of the meaning of Twitter data used by Eichstaedt et al., and these authors' complex analysis techniques (involving, for example, several steps to deal with high multicollinearity) are merely modeling this noise to produce the illusion of a psychological mechanism that acts at the level of people's county of residence. Certainly, the different levels of "negative" Twitter language between these two Indiana counties appear to have had no deleterious differential effect on local AHD mortality; indeed, at 45.4 deaths per 100,000 inhabitants, "angry" Jackson County's AHD mortality rate in 2009–2010 was 23.7% *lower* than "laid-back" Clay County's (59.5 per 100,000 inhabitants). As we showed in our analysis of Eichstaedt et al.'s comparative maps, this failure of Twitter language to predict AHD mortality with any reliability is widespread.

In a recent critique, *Jensen (2017)* examined the claims made by *Mitchell et al. (2013)* regarding the ability of Twitter data to predict happiness. Jensen argued that "the extent of overlap between individuals' online and offline behavior and psychology has not been well established, but there is certainly reason to suspect that a gap exists between reported and actual behavior" (p. 2). Jensen went on to raise a number of other points about the use of Twitter's "garden hose" dataset that appear to be equally applicable to *Eichstaedt et al. (2015a)*, concluding that "When researchers approach a data set, they need to understand and publicly account for not only the limits of the data set, but also the limits of which questions they can ask . . . and what interpretations are appropriate" (p. 6). It is worth noting that Mitchell et al. were attempting to predict happiness only among the people who were actually sending the tweets that they analyzed. While certainly not a trivial undertaking, this ought to be considerably less complex than Eichstaedt et al.'s attempt to predict the health of one part of the population from the tweets of an entirely separate part (cf. their comment on p. 166: "The people tweeting are not the people dying"). Hence, it would appear likely that Jensen's conclusions—namely, that the limitations of secondary data analyses and the inherent noisiness of Twitter data meant that Mitchell et al.'s claims about their ability to predict happiness from tweets were not reliably supported by the evidence—would be even more applicable to Eichstaedt et al.'s study, unless these authors could show that they took steps to avoid the deficiencies of Mitchell et al. On a related theme, *Robinson-Garcia et al. (2017)* warned that bots, or humans tweeting like bots, represent a considerable challenge to the interpretability of Twitter data; this theme has become particularly salient in view of recent claims that a substantial proportion of the content on Twitter and other social media platforms may not represent the spontaneous output of independent humans (*Varol et al., 2017*).

The principal theoretical claim of *Eichstaedt et al.*'s (*2015a*) article appears to be that the best explanation for the associations that were observed between county-level Twitter language and AHD mortality is some geographically-localized psychological factor, shared

[10] *Eichstaedt et al. (2015a)* included a disclaimer about causality on p. 166 of their article. However, we feel that this did not adequately compensate for some of their language elsewhere in the article, such as "Local communities *create* [emphasis added] physical and social environments that *influence* [emphasis added] the behaviors, stress experiences, and health of their residents" (p. 166; both of the italicized words here seem to us to imply causation at least as strongly as our word "exerts"), and "Our approach . . . could bring researchers closer to understanding the community-level psychological factors that are important for the cardiovascular health of communities and should become the focus of intervention" (p. 166, seemingly implying that an intervention to change these psychological factors would be expected to lead to a change in cardiovascular health).

[11] For example, using data from the CDC for the 2009–2010 period, county-level mortality from assault is strongly correlated with county-level mortality from cancer ($r = .55$), but completely uncorrelated with county-level mortality from AHD ($r = .00$). There seems to be no obvious theoretical explanation for these results.

by the inhabitants of an area, that exerts[10] a substantial influence on aspects of human life as different as vocabulary choice on social media and arterial plaque accumulation, independently of other socioeconomic and demographic factors. However, Eichstaedt et al. did not provide any direct evidence for the existence of these purported community-level psychological characteristics, nor of how they might operate. Indeed, we have shown that the same techniques that predicted AHD mortality could equally well have been used to predict county-level suicide prevalence, with the difference that *higher* rates of self-harm seem to be associated with "positive" Twitter language. Of course, there is no suggestion that the study of the language used on Twitter by the inhabitants of any particular county has any real predictive value for the local suicide rate; we believe that such associations are likely to be the entirely spurious results of imperfect measurements and chance factors, and to use Twitter data to predict which areas might be about to experience higher suicide rates is likely to prove extremely inaccurate (and perhaps ethically questionable as well). We believe that it is up to Eichstaedt et al. to show convincingly why these same considerations do not apply to their analyses of AHD mortality; as it stands, their article does not do this. Taken in conjunction with the pitfalls (*Westfall & Yarkoni, 2016*) of including imperfectly-measured covariates (such as Eichstaedt et al.'s county-level measures of smoking and health status, as described above) in regression models, and the likely presence of numerous substantial but meaningless correlations in any data set of this type (the "crud factor"; *Meehl, 1990*[11]), it seems entirely possible that Eichstaedt et al.'s conclusions might be no more than the result of fitting a model to noise.

## CONCLUSIONS

It appears that the existence of community-level psychological characteristics—and their presumed valence, either being "protective" or "risk" factors for AHD mortality—was inferred by *Eichstaedt et al. (2015a)* from the rejection of a series of statistical null hypotheses which, though not explicitly formulated by these authors, appear to be of the form "There is no association between the use of (a 'positive' or 'negative' language element) by the Twitter users who live in a given county, and AHD-related mortality among the general population of that county." Yet the rejection of a statistical null hypothesis cannot in itself justify the acceptance of any particular alternative hypothesis (*Dienes, 2008*)—especially one as vaguely specified as the existence of Eichstaedt et al.'s purported county-level psychological characteristics that operate via some unspecified mechanism—in the absence of any coherent theoretical explanation. Indeed, it seems to us that Eichstaedt et al.'s results could probably equally well be used to justify the claim that the relation between Twitter language and AHD mortality is being driven by county-level variations in almost any phenomenon imaginable. To introduce a new psychological construct without a clear definition, and whose very existence has only been inferred from a correlational study—as Eichstaedt et al. did—is a very risky undertaking indeed.

## ACKNOWLEDGEMENTS

We thank Casper Albers, Daniël Lakens, and a number of colleagues who wished to remain anonymous for helpful discussions during the writing of this article. All errors and omissions remain the responsibility of the authors alone.

### Funding

The authors received no funding for this work.

### Competing Interests

James C. Coyne is an Academic Editor for PeerJ.

### Author Contributions

- Nicholas J.L. Brown conceived and designed the experiments, performed the experiments, analyzed the data, contributed reagents/materials/analysis tools, prepared figures and/or tables, authored or reviewed drafts of the paper, approved the final draft.
- James C. Coyne authored or reviewed drafts of the paper, approved the final draft.

### Data Availability

Brown, Nicholas JL. 2018. "Reanalysis of Eichstaedt et al. (2015: Twitter/heart disease)." Open Science Framework. February 12. https://osf.io/g42dw/.

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
