# Peer review of "Does Twitter language reliably predict heart disease? A commentary on Eichstaedt et al. (2015a)"

_PeerJ, doi:10.7717/peerj.5656_

## Round 0.1 · original submission · Major Revisions

It is PeerJ policy that in cases where a specific article is being critiqued we ask the authors to also provide a review. We have therefore invited Johannes Eichstaedt to review the manuscript and asked him to sign the review in the interest of transparency.

It is important that published studies are checked critically, and the manuscript has the potential to make an important contribution to the field, as also reflected by the comments of the reviewers. However, two main areas need to be revised before the manuscript can be considered for publication.

1) The relation to the original paper should be clarified. The manuscript comments on the data that was used, the analysis of this data and presents additional analyses. Eichstaedt et al. (2015) used data rather than acquiring the data themselves. Although the concerns on the reliability of this data are important, they do not specifically apply to the original article but to the field in general and should hence be discuss accordingly. The assessment of the analyses in Eichstaedt et al. (2015) is most relevant to the aim of the current manuscript. Some concerns that are raised are quantitatively tested, such as the inclusion of ‘love’, and this is insightful as pointed out by reviewer 3, while other concerns are not tested, e.g. inclusion of additional variables such as access to good-quality healthcare, socioeconomic status and Gini coefficient of income equality and the role of outliers. As the data and code is available these concerns should be quantitatively tested. If this is not possible because the relevant data is not available, this should be clearly stated and speculations should be kept to a minimum. The additional analyses that are presented (relation with suicide) may help to put the original findings into context, but the robustness of this finding to confounding variables needs to be addressed.

2) The writing and structure of the manuscript should be revised. The wording should be more factual and the original study should be accurately discussed. Assumptions on the intentions of original authors that are not based on the article (e.g. Type A personality) and insinuations (violation of Psychological Science standards) should be removed. Misrepresentations of the original text (e.g. causal interpretation) should be corrected. The authors should also try to minimize redundancies in the manuscript.

Since the authors submitted their manuscript to PeerJ, the original authors have posted a preprint (http://doi.org/10.17605/OSF.IO/P75KU), responding to the critique in the present manuscript that was also posted as a preprint. As this preprint is highly relevant to the current manuscript, it should be cited to ensure the current manuscript is up-to-date when published at PeerJ. Although the authors do not need to provide a detailed response to all points raised in Eichstaedt et al. (2018), which I would advise against, they should revise their comments on the availability of data and code and address the role of confounding variables in the analysis of suicide.

Reviewer 1 ·

Basic reporting

The paper is well written and enjoyable to read. For the most part, the arguments in the paper are supported by theory and sufficient references. Some comments:

1. I think the paper overstates how much Americans move away during their lives, and doesn't provide sufficient citations when discussing this (lines 123, 310). For example, this article suggests that it is fairly rare to move far from home: https://www.nytimes.com/interactive/2015/12/24/upshot/24up-family.html (sorry I don't have an academic reference, but the article links to the original data source).

2. The paper insinuates (lines 248+) that Psychological Science violated their standards by giving the award to a project that did not share their code, but this is not plainly true given only these quotes. It can be argued that a paper can provide enough detail to be reproducible without sharing their code, so it would strengthen this argument to state that the Eichstaedt article was unclear without code, and/or to cite a perspective that code is necessary (for example, Gundersen and Kjensmo (AAAI 2018) define code sharing as a necessary component of reproducibility in computational research).

3. Line 177: it's not obvious why misspellings would invalidate the use of these words as a measurement.

Experimental design

The methods are well justified, following Eichstaedt et al. as closely as possible.

One comment:

1. I was fascinated by the discussion of the variability of how AHD is measured. It's a strong point to make. It made me wonder (I don't have the domain expertise to answer myself), are there other ICD codes besides I25.1 that would capture similar conditions? This study used a range of codes for suicide, so I wonder if a range of codes could similarly be used for cardio causes of death.

Validity of the findings

There is at least at least one place where the article states that the problems of Eichstaedt et al. apply to all social media research. (544: "It appears that we are some way from being able to generate reliable estimates of health outcomes based on tweets.") Of course, this article does not go far enough to make this claim; a lot of the issues are specific to this study.

For example, detecting flu at the national level from tweets is a very different problem, which relies on a straightforward observable construct, and does not have the issues of counties/communities. This article does cite the failure of Google Flu Trends, but those problems were due to poor practices on the part of Google, and do not necessarily apply to newer systems that have been more carefully validated (see e.g. Santillana, PLOS CompBio 2015, e1004513).

If the authors do want to make generalizations to other social media research, a more helpful way to do it would be to add discussion about what lessons can be learned from this analysis to be applied to social media research more broadly. While I can infer interesting takeaways throughout the analysis, it would strengthen the paper to make them more explicit.

Additional comments

Overall, I think this paper makes a strong argument that the findings of Eichstaedt et al. have been misinterpreted. There are interesting lessons to be learned here for using social media to study health outcomes. I think the paper could do a better job of making these lessons explicit, and I have a few suggestions for improvement in my specific comments.

Reviewer 2 ·

Basic reporting

The two authors put the claims in Eichstaedt et al. 2015's work to test. In particular, they test whether there is a relationship between the use of language in tweets and US county-level mortality rates from atherosclerotic heart disease. I commend the authors for their extensive work in verifying the claims made in Eichstaedt et al.'s article. The manuscript's topic is timely, its methodology is robust, and the results are compelling. Also, the manuscript is clearly written, yet a few sentences should be rephrased to avoid reporting personal opinions, jokes and ironic statements.

Experimental design

The design is robust. One small thing: the choice of "mortality rates" as outcome variable should be further motivated.

Validity of the findings

The findings are valid.

Additional comments

Please check the following statements and modify them accordingly:

- 127, pls add citations supporting the following sentence, "there is another caused of death for which the existence of an association with the victim's immediate psychological state is widely accepted, namely suicide."

- 149, please clarify what you mean by randomly and why the random nature of the selection is important here, "randomly-selected tweets"

- 231, replace "apparently chose not to include " with "did not include"

- 363, i would remove "before any program .. to be undertaken." no need for it unless Eichstaedt et al.'s work states that as the main aim of their study

- 472-73, remove "as they might have preferred". you don't know that.

- 559, remove "unless the transmission of this characteristic ... telepathic level." it s unnecessary

- 608, rephrase "slight wrinkle", irony does not contribute to this manuscript's quality (on the contrary!)

- 636-7, remove "variations in chakra energy or UFO activity"

·

Basic reporting

The manuscript is well written in clear, unambiguous, professional English language. Intro and background show the context, and the literature is well referenced and relevant. The structure of the manuscript is also clear, figures and tables are relevant. Raw data are supplied. Indeed, the reanalysis is extremely transparent, all data and scripts can be found on OSF, supplemented with a very helpful code and data documentation. I partially checked the raw data and scripts and could easily reproduce the reported results.

However, when I read the manuscript, I also observed that some issues appeared twice in the manuscript, once in the introduction and once in the results section. I had the feeling that this might not be the easiest structure for a reader to follow, and that it might be good to restructure your manuscript a bit to increase its accessibility.

The current structure is
A) Introduction
A1) General Introduction (lines 36-63)
A2) Problems that occurred from reading the original article (lines 64-241)
B) Method (lines 242-166)
C) Results: Problems that occurred from the reanalysis (lines 268-528)
D) Discussion (lines 529-622)
E) Conclusion (lines 623-639)

In my view, the problem is the separation of part A1 (Problems that occurred from reading the original article without looking in the data) from part C (Results: Problems that occurred during the reanalysis). This separation has the disadvantage that many of the issues appear twice in the manuscript (usually, the second time with an introducing sentence such as “As previously noted…”, “We noted earlier…”, etc.). I think that it would be easier to follow for the reader when each issue would be handled only once, and I would thus suggest that you move some of the parts of A2 to the Results section.

Specifically, I suggest moving lines 100-112 of the part “issues related to the etiology of AHD” to the beginning of the part “variability in ICD-10 coding of cause of death” (lines 282ff.), because both sections deal with the unreliability of the physician’s diagnosis of AHD mortality. Lines 113-133 of this part would then fit to the beginning of the part “Use of mortality from AHD as the outcome variables” (lines 307ff.), because then the introduction of the idea of analyzing suicides and the actual analysis of suicides comes together in one section. The part “issues associated with considering counties as communities” (lines 186ff.) would well fit to the part in the results section titled “problems associated with considering counties as communities” (lines 386ff.). As you even see from the titles, both parts deal with exactly the same topic, and I think that it would be easier for the reader when you provide your critique of counties that is independent from a re-analysis together with the critique that depends on your re-analysis. The “possible sources of bias in the geographical coding of Twitter data” (lines 151ff.) and the “potential source of bias due to derivation of topics with Facebook” (lines 169ff.) could similarly go together with the part “bias caused by selection of counties” (lines 365ff.). For the reader, it does not make a big difference whether you observed a bias by carefully looking at the supplement or by looking at the data.

I want to end this by saying that this re-structuring is just an idea and that there might be very good reasons why you chose the current structure. I just had the feeling that your manuscript would be more accessible with a changed structure, but it is absolutely fine with me if you keep the structure as it is.

Experimental design

The research is within the scope of the journal, as it deals with an important health issue. The research question is well defined, relevant and meaningful. The analysis was performed rigorously to as high technical standard. The analyses are described in sufficient detail and all information to reproduce the results is provided.

Validity of the findings

Data is robust, statistically sound and controlled. Conclusions are well stated, linked to original research question and limited to supporting results. Speculation is identified as such.

Additional comments

I read your manuscript “No evidence that Twitter language reliably predicts heart disease: A reanalysis of Eichstaedt et al. (2015a)” with great interest. This manuscript reports a reanalysis and an extension of an original manuscript “Psychological language on Twitter predicts county-level heart disease mortality” by Eichstaedt et al. published in Psychological Science. The original article reports that the language used on Twitter, aggregated at the level of U.S. counties, predicts county-level mortality rates from atherosclerotic heart disease (AHD). Surprisingly, not only could the authors find evidence for the effect of language, but they could even show that Twitter language predicted AHD mortality significantly better than a model combining socioeconomic and health risk factors, including smoking, diabetes, hypertension, and obesity. This result has possible societal implications, which is mirrored by the fact, that the article has already been cited 173 times in only three years. One could be a bit skeptical about the findings in Eichstaedt et al. (2015), given that the two most important results are only significant at p = .026 (incremental validity of Twitter over and above traditional risk factors) or p = .049 (comparison of Twitter model to traditional risk factor model).

Indeed, you now identified in your manuscript a number of factors that question the result by Eichstaedt et al. (2015). At first, I would like to thank you for the work you put into the reanalysis of Eichstaedt et al.’s study. It is important for the field that published studies are checked critically. This is exactly how science should work, but which we do not see that often in psychological and medical research.

Taken together, your new analyses and arguments make me doubt that the original finding is as robust as Eichstaedt et al. (2015) stated in their original publication. For me, the most important points were the severe problems with the data that you identified. In particular, problems in the coding of AHD mortality, as shown by a range from 1.55% to 26.11% between counties, which is a far more extreme range in the main dependent variable than I would have expected – and which is not plausible to me at all. Furthermore, the extreme differences in population size between counties, the selection bias in counties, and the censoring of Twitter seem also all very relevant to me, but not mentioned in the original article. Furthermore, I found the results very interesting, when you included “love”, and when you excluded “hate” from the data. In fact, it is not really convincing to exclude “love”, but exclude “hate”, as Eichstaedt et al. did in their original analysis – and I agree with you that this decision was very likely based on the results.

As you might have seen already, Eichstaedt et al. responded to your critical points in a manuscript titled “More Evidence that Twitter Language Predicts Heart Disease: A Response and Replication” which was provided as a pre-print at OSF on March 15, i.e. during the time when your manuscript was under review. I think that it would be the best if both manuscripts were published so that everyone could read them and form his/her own opinion. Thus, I do not think that you should necessarily incorporate your response to Eichstaedt et al.’s response in the current manuscript, but that it might be better to write a new response to their response (of course, only if you like) and to publish the present manuscript as it is. This would be more transparent and the discourse would be easier to follow for the scientific community.

There is only one point that I think you could add to your manuscript in response to Eichstaedt et al.’s pre-print. You write in your manuscript that Eichstaedt et al. (2015) did not provide data and script for the prediction model. Instead, you even had to reconstruct the results for the prediction model from the figures using the different colors. In the meantime – in response to your paper - Eichstaedt et al. now re-released the data on the OSF in – as they write – more accessible form, ready for database import, and they provided in Appendix A of their response the code for the reproduction of the original findings of the prediction model.

I think that it would be good to add a footnote to your manuscript that the data and scripts for the ridge regression had been made available while the manuscript was under review, but that you did not use these data for writing your manuscript, because data and code were not available at that time. Alternatively, you could also have a closer look at the actual prediction model analysis and include this as a new section in your manuscript (but you could also leave this part for a possible response).

·

Basic reporting

## Note about this review

Note: The authors had released a version of this critique as a preprint on February 12th, 2018 (10.17605/OSF.IO/DURSW, version 2). The original author team of Eichstaedt et al. (which I lead; 2015) produced a substantive response which was released on March 15th, 2018 as a preprint (10.17605/OSF.IO/P75KU, version 2; Eichstaedt et al., 2018). This response also includes a detailed rebuttal of most points made by Brown and Coyle (see Appendix B). This was concluded before I accepted the invitation to serve as a reviewer (March 22nd, 2018). In agreement with the editor, I will reference this response throughout this review, as it contains a substantive (and we hope, even-handed) analysis Brown and Coyne’s critique.

Eichstaedt, J. C., Schwartz, H. A., Giorgi, S., Kern, M. L., Park, G.,... Ungar, L. H. (2018, March 15). More Evidence that Twitter Language Predicts Heart Disease: A Response and Replication. http://doi.org/10.17605/OSF.IO/P75KU. https://psyarxiv.com/p75ku/.

Fundamentally, their critique is not a reanalysis of our work, contrary to the title of the manuscript, and the claims made in the abstract. The authors’ critique does not attempt a replication of our claim, implied in their title, that county-level Twitter language predicts county-level heart disease rates.

Instead, it contains a different analysis of language correlates of county-level suicide rates, which the authors claim should match the county-level correlates of heart disease mortality. In our response (Eichstaedt et al., 2018), we found suicide rates to be are uncorrelated with heart disease rates and their linguistic correlates to mostly disappear when county elevation and rural populations are controlled for (unlike heart disease associations), suggesting that county-level suicide rates are not a straightforward measure of county-level psychological health. A CDC-reported measure of poor mental health based on phone surveys, on the other hand, shows the same pattern of correlations with psychological Twitter Language as does heart disease mortality (see Eichstaedt et al., 2018).

The manuscript would be better framed as related work, or developed into an even-handed commentary on social media-based or epidemiological methods more generally. More details are given below.

The most important limitations are flagged in the experimental designs section.

## Concerns about basic reporting -- Summary

In terms of basic reporting, I would like to see the following shortcomings addressed before a possible publication of the manuscript. (a) Claims made in Eichstaedt et al., 2015 are misrepresented; (b) the discussion of the sources of noise fails to acknowledge the principle claim of Eichstaedt et al., 2015, namely, that prediction models can use Twitter language encodings to predict heart disease mortality out-of-sample, that is, in an experimental set up in which sources of noise would work against (and not for) the ability to predict heart disease rates; (c) concerns about different kinds of noise in county-level variables and Twitter data are speculative and one-sided, lack empirical estimates, and do not properly acknowledge the large literature using these data sources.

## (a) Misrepresentations of the original article

There are a number of misrepresentations in the manuscript, which are identified in detail in Appendix B of Eichstaedt et al., 2018. The most severe ones are flagged here.

# Claims of independent psychological causation

The authors allege that:

“The principal theoretical claim of Eichstaedt et al.’s (2015a) article appears to be that the best explanation for the associations that were observed between county-level Twitter language and AHD mortality is some geographically-localized psychological factor, shared by the inhabitants of an area, that exerts a substantial influence on aspects of human life as different as vocabulary choice on social media and arterial plaque accumulation, independently of other socioeconomic and demographic factors.” (p. 26, lines 599ff)

The 2015 article did do not claim independence of psychological markers from socioeconomic and demographic factors (and the principal claim was stated in our title, “Psychological Language on Twitter Predict County-level Heart Disease Mortality”). Regarding the authors’ claim, the article stated at the end of the results section:

“Taken together, these results suggest that the AHD relevant variance in the 10 predictors overlaps with the AHD-relevant variance in the Twitter language features. Twitter language may therefore be a marker for these variables and in addition may have incremental predictive validity.” (p. 164)

The article also specifically disclaimed making causal inferences, as it was purely cross-sectional:

“Finally, associations between language and mortality do not point to causality; analyses of language on social media may complement other epidemiological methods, but the limits of causal inferences from observational studies have been repeatedly noted (e.g., Diez Roux & Mair, 2010)." (p. 166)

Please clearly state the claims made by the 2015 article throughout the manuscript – psychological markers measured through Twitter mark, and in large part overlap in variance, with classical predictors, among them predominantly income and education (see Figure 2 in Eichstaedt et al., 2015).

# Type A personality

The authors suggest in their introduction (p. 4) that the claims made in the 2015 article hinge on the empirical adequacy of Type A personality theory, and then continue to discuss the mixed findings that Type A personality theory may have received. Our 2015 article did not mention Type A personality theory – it observed a set of correlations at the community level, and, in the discussion, compared them to what has been observed at the individual level. Specifically, it referenced individual-level associations with depressed mood, anger and anxiety:

“Our findings point to a community-level psychological risk profile similar to risk profiles that have been observed at the individual level. County-level associations between AHD mortality and use of negative-emotion words (relative risk, 5 or RR, = 1.22), anger words (RR = 1.41), and anxiety words (RR = 1.11) were comparable to individual-level meta-analytic effect sizes for the association between AHD mortality and depressed mood (RR = 1.49; Rugulies, 2002), anger (RR = 1.22; Chida & Steptoe, 2009), and anxiety (RR = 1.48; Roest, Martens, de Jonge, & Denollet, 2010).” (p. 164)

The authors should properly represent the claims made in the 2015 article (about hostility, depressed mood, anger and anxiety), and not suggest that the analysis is based on Type A personality theory, or require any specific personality theory to be correct.

# Unavailability of data

The authors claim that “However, as far as we have been able to establish, Eichstaedt et al. did not provide any of the code needed to reproduce their analyses (…)” (e.g., page 11)

The 2015 author team released both the county-level (a) Twitter language and (b) outcome data in a way that allowed people with some effort to reproduce the 2015 findings (county-level topic, dictionary, and 1-to-3-gram frequencies, see https://osf.io/rt6w2/). An early version of the code base was released on the research group’s homepage (wwbp.org) later in 2015. Since then, usability and documentation has been improved and it has been published and released open source in 2017 (Differential Language Analysis ToolKit, dlatk.wwbp.org; Schwartz et al, 2017). Additional step-by-step instructions to reproduce the original prediction accuracies can be found in Eichstaedt et al., 2018, Appendix A.

The authors should acknowledge that the code has be released publicly since the publication (in both 2015 and 2017), and that they made no attempt to contact the 2015 author team, who could have easily pointed them to these resources (as they already have to others).

## (b) Misunderstanding about the role of noise/bias in out-of-sample Evaluation

The manuscript in its current form fails to acknowledge the fact that the principal claim that Twitter language encodings can be used to predict heart disease rates are determined out-of-sample, that is, in an experimental set up in which the prediction models are only evaluated on “test” counties not used during model fit (“training”). In other words, in our work, language patterns that generalize across both training and test counties contribute to correct predictions – and sources of noise in the data would likely degrade the ability of the models to find patterns that generalize across all counties.

The current discussion of sources of error, bias and noise in the manuscript (including the unrepresentativeness of Twitter users, Twitter bots, county conditions changing over time, potential unreliability of death certificates, etc.) is framed rhetorically to suggest that the possible sources of error make our finding that Twitter predicts AHD less reliable or robust. But these sources of noise no not disagree in substance with the results presented in the 2015 article, despite a rhetorical framing to the contrary used throughout the manuscript.

Instead, the 2015 article establishes empirically how much of the variance in heart disease can be predicted *despite* all these sources of error (which are mostly acknowledged in the original manuscript). In other words, the manuscript lists possible reasons why the out-of-sample prediction accuracies reported in the 2015 article are as relatively low as they are (out of sample accuracy r = 0.42) – accounting for 17% of the variance in heart disease--but offers no substantive critique of the methods to create these predictions, nor attempts a replication.

Please properly state the effect of the possible sources of noise/bias/error in light of the out-of-sample evaluation methods used in the 2015 article, and contextualize them in light of the relatively modest claims about how much of the variance in heart disease can be predicted used Twitter language. And please remove suggestions that the 2015 original analysis either claimed or claimed “implicitly” (line 161, p. 8) that the Twitter data sources are representative.

## (c) Speculative & one-sided critiques

The authors note that there are various sources of noise in the geo-located Twitter data and the county-level outcome data. In their role as a refutation of the claims made by the 2015 article, they are insufficiently contextualized considering the out-of-sample evaluation used in the analysis (see section (b) above).

As a commentary on social-media based and spatial / epidemiological methods more generally they lack (1) empirical estimates or estimates of the relative importance in their effect on these kinds of analysis, and more importantly, (2) an even-handed treatment of the large literature that uses these kinds of data, spanning computer science, public health and epidemiology among others. This includes critiques of big data and Twitter methods that have been published previously.

# …Regarding Twitter Data

For example, as was noted in the 2015 article, users who tweet are not representatively selected, and some of the tweets (7%) are incorrectly mapped to counties. Further, some people may move from county to county, the way the “Garden Hose” Twitter sample is selected is non-random or otherwise imperfectly provided by Twitter, there are bots on Twitter, etc. These critiques are valuable, but in their current form are insufficiently integrated with the rest of the literature. A large number of publications has used geo-tagged Twitter data for all sorts of applications including health predictions—the 2015 article is far from the only article using these methods.

e.g.

Culotta, A. (2014, April). Estimating county health statistics with twitter. In Proceedings of the SIGCHI Conference on Human Factors in Computing Systems (pp. 1335-1344). ACM.

Or, for critical reviews of these methods consider:

Pavalanathan, U., & Eisenstein, J. (2015). Confounds and consequences in geotagged Twitter data. arXiv preprint arXiv:1506.02275. (Published at EMNLP)

If a critical appraisal of social media-based methods is included, please provide a more balanced survey of the use and discussion of geo-tagged Twitter data in current research.

# …Regarding heart disease data

The authors doubt that the coding on death certificates for underlying cause of death is reliable (p. 13).

We agree that, like outcomes used in nearly all public health studies, there is some degree of error in the heart disease mortality rates and other outcome variables, and we noted this in the original paper. The source of mortality data we used (the mortality rates from the Centers for Disease Control and Prevention’s Wide-ranging Online Data for Epidemiologic Research database, or CDC Wonder for short) are widely used in research. Our analysis and hundreds of others do in fact depend on the assumption the main source of variance within these officially-reported data to be what they profess to measure, in the same way that these outcomes and estimations are used throughout medical and public health research.

e.g.

Pinner, R. W., Teutsch, S. M., Simonsen, L., Klug, L. A., Graber, J. M., Clarke, M. J., & Berkelman, R. L. (1996). Trends in infectious diseases mortality in the United States. JAMA, 275(3), 189-193.

Jemal, A., Ward, E., Hao, Y., & Thun, M. (2005). Trends in the leading causes of death in the United States, 1970-2002. JAMA, 294(10), 1255-1259.

Murray, C. J., Kulkarni, S. C., Michaud, C., Tomijima, N., Bulzacchelli, M. T., Iandiorio, T. J., & Ezzati, M. (2006). Eight Americas: investigating mortality disparities across races, counties, and race-counties in the United States. PLoS medicine, 3(9), e260.

Hansen, V., Oren, E., Dennis, L. K., & Brown, H. E. (2016). Infectious disease mortality trends in the United States, 1980-2014. JAMA, 316(20), 2149-2151.


In addition, the McAllum, St. George and White (2005) citation given by the authors to support the unreliability of heart disease death statistics (line 306, p. 13) is based on the qualitative output of four teleconferenced focus groups across 16 General Practitioners in New Zealand, which concludes that “Improving death certification accuracy is a complex issue.” This appears to have limited applicability to CDC-reported heart disease rates in the U.S., which the 2015 article is based on.

Please acknowledge the wide use of these data sources in health research, and if a critique of the use of such data sets in epidemiological research is intended, please provide an even-handed accounting of the reliability of these data in previous research.

## Referencing

A number of citations given throughout the article do not appear to support the claims after which they are provided (these are relatively minor points).

//

Page. 4, line 80 -- “At best, negative affectivity is likely to be no more than a non-informative risk marker (Ormel et al., 2004), not a risk factor for AHD. Its apparent predictive power is greatly diminished with better specification and measurement of confounds (Smith, 2011).”

Both of these references are about personality traits, which are not referenced in our paper.

Ormel, J., Rosmalen, J., & Farmer, A. (2004). Neuroticism: A non-informative marker of vulnerability to psychopathology. Social Psychiatry and Psychiatric Epidemiology, 39, 906–912.

Smith, T. W. (2011). Toward a more systematic, cumulative, and applicable science of personality and health: lessons from type D personality. Psychosomatic Medicine, 73, 756 528–532.

The Ormel, Rosmalen and Farmer paper critiques the idea of neuroticism (NOT negative affectivity) as being informative as an explanatory construct over and above it describing a characteristic level of distress (which it acknowledges is connected to health outcomes).

The Smith article is an editorial comment about Neuroticism and Type D Personality. The 2015 article makes no claims about personality.

//

Page 13, line 306 -- McAllum, St. George and White, 2005 – see above.

//

Page 22, line 496

“thus the statistical power to detect an effect sufficiently high, that these p values arguably constitute quite strong evidence in favor of the null hypothesis of no effect (cf. Vazire, 2017).”

While I am sympathetic to the statistical idea being conveyed here, the reference given is to a non-peer-reviewed blog post titled “be your own a**hole.”

Experimental design

## Main Analyses

The manuscript claims to be a re-analysis of the 2015 Eichstaedt et al. paper in abstract and title. The 2015 paper had two major sections: (a) an evaluation of Twitter to cross-sectionally predict heart disease mortality, and (b) an exploration of the language correlates of heart disease mortality. This manuscript attempts a re-analysis of neither and thus its framing is inappropriate.

The main empirical contribution of the manuscript is the exploration of the Twitter language correlates of county-level suicide rates, which is not a reanalysis of the Eichstaedt et al. (2015) study. I will evaluate in this section (1) the quality of the statistical analyses, and (2) how the statistical analyses presented relate to the claims made in the article title and abstract.

Regarding (1), the reported dictionary correlations with age-adjusted suicide rates from CDC Wonder across the N = 741 counties for which sufficient data were available reported in Table 1 appear correct, and have been independently reproduced in Eichstaedt et al., 2018 (compare Table 1 in this manuscript with Table 1 in Eichstaedt et al., 2018). Given the high intercorrelations between dictionary and topic frequencies (see Table S2 in Eichstaedt et al., 2018), the topic-based findings visualized in Figure 2 also appear plausible.

However, I remain wholly unconvinced about (2), namely that the statistical analysis about suicide rates presented in the manuscript are meaningfully related to the heart disease correlations presented in the 2015 article.

Specifically, the authors connect the suicide analysis to the heart disease analysis through the following theoretical assumption: “we might expect county-level psychological factors that act directly on the health and welfare of members of the local community to be more closely reflected in the mortality statistics for suicide than those for a chronic disease such as AHD.” (page 6).

However plausible this may appear at the individual level, no empirical support is given for this critical assumption at the aggregate, county level, nor is the literature on the epidemiology of suicides properly considered.

In fact, the literature suggests suicides are a complex (and often baffling) mortality outcome that shows strong and robust links at the county level to (a) elevation (r = .51, as reported by Kim et al., 2011, and r = .50, as reported by Brenner et al., 2011) (perhaps because of the influence of hypoxia on serotonin metabolism; Bach et al., 2014) in addition to (b) living in a rural areas (e.g., see, Hirsch, 2006, for a review; Searles et al., 2014), attributed in part to social isolation and insufficient social integration, a trend that has increased over time (Singh & Siahpush, 2002; see Eichstaedt et al., 2018 for a full discussion). Implicit in these associations is also the increased availability of guns in rural communities, the preferred means of committing suicide.

In the original 2015 analysis, we tested the dictionary associations with heart disease rates using its strongest covariates as controls (income and education), finding that most of the negative language variables remained significantly associated.

In the response to this manuscript (Eichstaedt et al., 2018), we focused on an analysis of dictionary correlations for parsimony. When controlling the associations between Twitter dictionaries and suicide rates for its strongest covariates--elevation and rural population--the language correlations reported in this manuscript are no longer significant (see Eichstaedt et al., 2018, Table 1, column 2). In contrast, controlling the association between heart disease rates with Twitter dictionaries for elevation and rural population does not noticeably affect those coefficients – all significant associations remain significant (see Eichstaedt et al., 2018, Table 1, columns 3 and 4).

This suggests that the associations reported for county-level suicide rates are not robust, and likely in large part driven by county-level confounding variables.

To more directly test the authors’ hypothesis that more psychological variables ought to be better candidates for association with psychological Twitter language, we tested the most psychological variable we had released with the original 2015 paper: the number of mentally unhealthy days people reported on average in a county, based on the CDC’s Behavioral Risk Factor Surveillance System (BRFSS). Unlike suicides, mentally unhealthy days correlate with the psychological dictionaries in the same directions and roughly at the same magnitudes as heart disease mortality (see Table 2 in Eichstaedt et al, 2018).

To summarize, a) the suicide language correlations are not robust and likely associated with confounds which are well known to the suicide literature, b) a clear CDC-reported measure of county psychological health shows the same correlations as does heart disease mortality.

Thus, in my view the empirical analysis offered in this manuscript is insufficiently developed to support the claims made in the title and abstract—that is, the suicide analyses are unrelated in important ways to the heart disease mortality and unsuitable as empirical support offered in a manuscript that is framed as a “re-analysis” of the 2015 heart disease article.

As its own contribution to the literature on the epidemiology of suicide, it would require further development and integration with the existing literature.

# Spatial analyses

The authors split the U.S. into a Northern and a Southern half and report language associations to appear stronger in the North but mostly disappear in the Southern Half. This raises a pertinent research question about how psychological phenomena should be treated statistically under spatial aggregation.

Geographic psychology is a relatively new subfield within psychology, and the authors raise the question about what role specialized regression techniques from other fields (like spatial regressions from geography) may play in improving our ability to robustly relate psychological phenomena to different spaces. This analysis could be developed further, but should be clearly framed as an open research question in psychology that is relevant to most spatial analyses that span an area as wide and diverse as the U.S.

Validity of the findings

As stated in the experimental design section, the claims made in title and abstract are not supported by the suicide-related analyses in the manuscript. The conclusions drawn by the authors are not supported by the experimental design.

I see a number of directions in which this manuscript could be developed.

(a) It could be reframed as an exploration of the language correlates of suicides. In that case, the analyses need to be properly developed to incorporate what is known about county-level suicide rates. In the current form, the analyses appear insufficiently developed.

(b) The manuscript could be developed into a full commentary of social-media based and/or epidemiological methods used in psychology. In that case, an even-handed review of the associated literatures would be required (in its current form it appears one-sided and largely speculative).

(c) The analyses could be replaced with an evaluation of the correlational profiles of a wider variety of more unambiguously psychological variables and considered in light of the associated literatures. To the extent such a manuscript is still framed as a commentary on the heart disease work, a clear empirical rationale ought to be developed how the analyses relate to the heart disease results.

(d) The manuscript could attempt a replication of the results presented by Eichstaedt et al., 2015 (see Eichstaedt et al., 2018, Appendix A for step-by-step instructions) and report on it.

(e) The manuscript could explore issues of spatial aggregation of psychological phenomena.

In the current form of the manuscript, none of these issues is sufficiently developed.

Additional comments

To reiterate, this critique is ill-framed as a re-analysis of the Eichstaedt et al., 2015 article, and it does not, in fact, attempt a replication of the key claim of that article (that county-level Twitter language predicts county-level heart disease rates). Instead, it contains an exploratory analysis of language correlates of county-level suicide rates--which are (a) uncorrelated with heart disease rates and (b) disappear when county elevation and rural populations are controlled for, suggesting that county-level suicide rates are not a straightforward measure of county-level psychological health. In addition, (c) a CDC-reported measure of poor mental health based on phone surveys, on the other hand, shows the same pattern of correlations with psychological Twitter Language as does heart disease mortality.

The manuscript contains a number of speculative doubts about data and methods that do not just apply to the Eichstaedt et al., 2015 paper, but to large literatures in computer science, public health and epidemiology, which are insufficiently taken into account in this manuscript.

---

## Round 0.2 · Major Revisions

The revised manuscript does not meet the criteria for publication in PeerJ. The primary concern is that the conclusions are not supported by the available results. In the original paper, Eichstaedt et al 2015 used a cross-validation process and compared their model using Twitter data against an alternative model based on relevant demographic data. In the current manuscript, several concerns are raised but the main results are an association between “self-harm” and Twitter language usage, showing a correlation between self-harm and negative language that remained significant after adjusting for income and education. However, the current study does not use a cross-validation approach and Eichenstaedt et al 2018 showed that this association is no longer significant when controlling for two other demographics (elevation and rural population). Taken together, these results suggest that the evidence is stronger that Twitter data is associated with heart disease mortality (as presented in Eichstaedt 2015) than with self-harm (as presented in the current manuscript). Hence, the main conclusion as reflected in the title, abstract and discussion are not justified in light of these results.

The authors should revise the manuscript and change the conclusions to make them consistent with the available results, including those presented in Eichstaedt et al 2015 and 2018 which pertain to the same data as used in the current manuscript. The authors should also address all reviewer comments. Please make sure that all responses to issues in the rebuttal document are also incorporated into the revised manuscript, as it is very likely that future readers will have similar questions. I will review the revisions myself and make the final decision whether the revised manuscript meets the criteria for publication in PeerJ.

Reviewer 1 ·

Basic reporting

The authors adequately responded to the general feedback and to my specific comments in the previous round (Review 1).

However, there seems to be a misunderstanding of the work in the comment responding to point #3 in my original review regarding misspellings. The "topics" are not manually curated by Eichstaedt et al., but are identified automatically with clustering algorithms (that's in fact a big part of the 'open vocabulary' approach they use in their work). Misspelled words are thus not misspellings on the part of Eichstaedt et al., but rather misspellings of Twitter users that were present in the data. Unless there is a misunderstanding on my part, I suggest the authors double check their understanding of how the 'topics' are interpreted, and make sure their manuscript describes them accurately.

Experimental design

The authors addressed this adequately.

Validity of the findings

The authors addressed this adequately.

Additional comments

No additional comments.

·

Basic reporting

No comment.

Experimental design

No comment.

Validity of the findings

No comment.

Additional comments

I thank you for the thorough revision and the detailed explanations in the cover letter. I have no further questions. I think that this manuscript makes an important contribution, and I am looking forward to seeing this paper being published.

·

Basic reporting

(see attached PDF document)

Experimental design

(see attached PDF document)

Validity of the findings

(see attached PDF document)

Additional comments

(see attached PDF document)

---

## Round 0.3 · accepted · Accept

The authors have addressed the main concerns and changed the main conclusion and title in line with the reviewer and editorial comments such that these are supported by the presented results. I am therefore happy to accept the manuscript for publication in PeerJ.

To get to this point there has been a lot of discussion conducted via the peer review reports and your subsequent revisions, however much of this discussion is only present in your rebuttal documents (and the peer review reports). I believe that having access to this discussion, and being able to view the peer-review process will be extremely valuable to your readers. PeerJ does have the ability to make the peer-review history fully public (the vast majority of their authors choose to do so) and so I would like to make it a condition of Acceptance that you also make the peer-review history public, so that readers can see the thought and process that has gone into this publication.